# Toward the Adoption of Anaerobic Digestion Technology through Low-Cost Biodigesters: A Case Study of Non-Centrifugal Cane Sugar Producers in Colombia

**Oscar Mendieta [1,2,\*], Liliana Castro [3,\*], Erik Vera [4], Jader Rodríguez [1] and Humberto Escalante [2]**

1 Corporación Colombiana de Investigación Agropecuaria-AGROSAVIA, Centro de Investigación Tibaitatá, 14 km Vía Mosquera Bogotá, Mosquera 250040, Colombia; jrodriguezc@agrosavia.co

2 Grupo de Investigación en Tecnologías de Valorización de Residuos y Fuentes Agrícolas e Industriales para la Sustentabilidad Energética-INTERFASE, Escuela de Ingeniería Química, Universidad Industrial de Santander—UIS, Carrera 27, Calle 9 Ciudad Universitaria, Bucaramanga 680001, Colombia; escala@uis.edu.co

3 Centro de Estudios e Investigaciones Ambientales-CEIAM, Escuela de Ingeniería Química, Universidad Industrial de Santander—UIS, Carrera 27, Calle 9 Ciudad Universitaria, Bucaramanga 680001, Colombia

4 Grupo de Investigación en Ciencia de Materiales Biológicos y Semiconductores-CIMBIOS, Escuela de Física, Universidad Industrial de Santander—UIS, Carrera 27, Calle 9 Ciudad Universitaria, Bucaramanga 680001, Colombia; erik.vera@correo.uis.edu.co

\* Correspondence: omendieta@agrosavia.co (O.M.); licasmol@saber.uis.edu.co (L.C.)

**Abstract:** Anaerobic digestion using low-cost biodigesters (LCB) is a promising alternative for Colombian producers of non-centrifugal cane sugar (NCS). Since the integration of anaerobic digestion technology in this agro-industry is novel, it is critical to understand the factors that affect the acceptance behavior of such technology by NCS producers to develop future policies that promote the adoption of sustainable energy alternatives. This study aimed to analyze NCS producers' behavioral intention to use LCB by utilizing an extended technology acceptance model (TAM). Data from a survey of 182 producers were used to evaluate the proposed model empirically. The extended TAM accounted for 78% of the variance in producers' behavioral intention to use LCB. Thus, LCB acceptability could be fairly precisely predicted on the basis of producers' intentions. This study's findings contribute to research on the TAM and provide a better understanding of the factors influencing NCS producers' behavioral intention to use LCB. Furthermore, this approach can assist policymakers at the local and global levels, given that NCS is produced in various developing countries worldwide.

**Keywords:** anaerobic digestion acceptance; structural equation model; energy policy; sustainable energy technology; rural development

## 1. Introduction

The traditional production of non-centrifugal cane sugar (NCS) from sugarcane is found in many developing countries [1]. For example, NCS production in Colombia is the country's second-largest agricultural sector, after coffee, with 220,000 ha of sugarcane cultivation. Over 350,000 families participate in this agro-industry, which generates 287,000 direct jobs and employs approximately 12% of the country's economically active rural population [2]. However, the NCS agro-industry has historically faced many challenges related to low agricultural and processing productivity, substandard product quality, and producer organization issues, all of which have hampered entry into new markets. The latter is reflected in the fact that a sizable proportion of producers and workers live in poverty. The sugarcane used in NCS production generates approximately 24.6% of its mass in organic waste, resulting in negative environmental impacts. This agro-industry generates 3.9 million tons of waste per year (according to Colombian production conditions). Crop residues are normally burned in the open air, or wastewater is dumped into bodies of water, resulting in odors, greenhouse

gas (GHG) emissions, and water and soil pollution [3]. The issue of finding alternatives to the waste generated by the agro-industry is currently being addressed.

Previous research has revealed that anaerobic digestion (AD) technology contributes to sustainability and a circular bioeconomy in different agro-industrial sectors [4,5]. Among alternative conversion methods, AD was shown to be the most sustainable biomass-to-energy technology for municipal waste management, with 34 indicators utilized within the context of three-dimensional sustainability (economic, environmental, and social) [6]. The increase in the number of agricultural biogas plants is a manifestation of the ongoing energy transition and an opportunity to achieve the objectives related to the implementation of the circular bioeconomy [7]. Specifically, recent research has shown that the main residues in NCS production (i.e., agricultural crop residues and sugarcane scum), when managed by AD, achieve synergy for bioenergy production [8]. A maximum methane yield of $0.276 \ \text{Nm}^3 \ \text{CH}_4 \cdot \text{kg}^{-1} \ \text{VS}_{\text{added}}$ was obtained for co-digestion, which constitutes an efficient form to take advantage of biomass. Subsequently, the feasibility of the AD process in low-cost biodigesters with a tubular configuration was determined, achieving a specific biogas production of $0.132 \ \text{m}^3 \cdot \text{kg}^{-1}$ VS with a methane content of 50.4%, and profitability indicators confirmed the economic viability of the technology for NCS producers [9]. Then, through a life-cycle analysis, the environmental sustainability of the AD technology integrated into the NCS production process was documented, highlighting the mitigation of the eutrophication impact categories up to 99% [10]. As a result, renewable fuel (biogas) and a biofertilizer (digestate) were obtained to assist the NCS sector in transitioning to sustainability and a circular bioeconomy.

Although the technical feasibility and the economic [9] and environmental [10] benefits of integrating low-cost biodigesters (LCB) into the NCS sector have been established, the producers' acceptance of this technology has not been assessed. The overwhelming majority of AD studies have been approached from a technical, environmental, and economic perspective. Nevertheless, social acceptance, which is a pillar of sustainability and circular bioeconomy for technology adoption, has not been considered. The social acceptance of sustainable energy production systems should be addressed before the implementation stage, which would help stakeholders create policies that help spread the technology. Understanding the fundamentals of the factors influencing NCS producers' acceptance of LCB will assist in making successful decisions to disseminate AD technology. Technology acceptance is considered a very important issue in the field of agriculture. Many theories and models have been used to accurately and systematically identify factors affecting innovation acceptance [11].

The technology acceptance model (TAM) and theory of planned behavior under perceived production risks were combined to analyze the factors affecting farmers' intentions to adopt information and communication technologies in intensive shrimp production in Vietnam [12]. The TAM has also been used to promote smart farming by Iran farmers [13], for which constructs were established that directly impact behavioral intention. On the other hand, a systematic review was carried out to adopt digital agricultural technologies to transform current agricultural systems toward sustainability, based on the diffusion of innovation theory (DIT) [14]. The constructs employed in the TAM, integrated with the perceived innovation characteristics of the DIT, provided an even more robust model than either of the two models alone [15,16]. TAM and DIT have been used in a variety of disciplines such as water resource management [17,18], sociology [19,20], and agriculture [21]. However, the TAM and DIT could be combined with additional variables to improve the model's predictive capacity. This could offer an approach toward the acceptance of anaerobic digestion technology/innovation. Despite the vast amount of research undertaken using the TAM in additional scientific fields, the literature has not explored its application in the waste management sector or, in particular, in AD technology. This study attempts to fill this gap by understanding the factors that affect the acceptance of AD technology by NCS producers in Colombia toward the adoption of LCB through an extended TAM. Overall, this study contributes to research on the TAM and, for the first time, to an under-

standing of NCS producers' LCB acceptance behavior. The TAM integrated with DIT, as well as two additional constructs (perceived self-efficacy and facilitating conditions) used in this work, is a more comprehensive model for analyzing LCB acceptance but has not been used in previous waste management studies. This empirical research supports the validity of this integrated model and constitutes an important contribution to adopting new sustainable energy technologies for developing countries. The findings of this study can help policymakers encourage NCS producers to use LCB and overcome some of the bottlenecks that arise during the implementation of such technology as a waste management tool and its contribution to the circular bioeconomy.

## 2. Materials and Methods

### 2.1. Extended Technology Acceptance Model

The technology acceptance model (TAM) is a popular model widely used in numerous studies on the acceptance and usage of information technologies [22]. The TAM determines technology acceptance (actual use) based on behavioral intention to use. Behavioral intention to use is influenced by attitude toward use and the direct and indirect effects of perceived usefulness and perceived ease of use. The degree to which an individual believes that using a particular system will improve their job performance is described as perceived usefulness. In contrast, perceived ease of use refers to how much a person thinks it would be free of effort to use the system. Both constructs, perceived usefulness and perceived ease of use, jointly affect the attitude toward use, while perceived ease of use directly impacts perceived usefulness.

The TAM model proposed by Venkatesh and Davis [23] eliminates the component attitude toward use (which previously mediated some of the impact of perceived usefulness and perceived ease of use). As with the original TAM, the actual use of technology is determined by behavioral intention to use. The latter is the focus of this study to predict AD technology acceptance. According to the modified Venkatesh and Davis [23] model, the following hypotheses were proposed:

**Hypotheses 1 (H1).** *Perceived usefulness has a direct effect on behavioral intention to use*.

**Hypotheses 2 (H2).** *Perceived ease of use has a direct effect on behavioral intention to use*.

**Hypotheses 3 (H3).** *Perceived ease of use has a direct effect on perceived usefulness*.

Rogers [24] proposed the DIT, which is now widely used to define and justify adopting technologies, ranging from agricultural tools to organizational developments. Five innovation characteristics are included in the DIT: relative advantage, compatibility, complexity, trialability, and observability. However, previous research has found that only relative advantage, compatibility, and complexity are consistently linked to innovation acceptance [25]. Perceived usefulness is similar to relative advantage, while perceived ease of use is similar to complexity. On the other hand, compatibility refers to how well an innovation fits into potential adopters' existing values, past experiences, and needs. Potential adopters will shy away from a new idea if it does not match society's traditions and values. The DIT is similar to the TAM in that it emphasizes the psychological and social influences on an individual's behavioral intention to adopt new technology [22]. According to this modified model, and taking compatibility as a construct in the TAM, the following hypotheses were proposed:

**Hypotheses 4 (H4).** *Compatibility has a direct effect on perceived usefulness.*

**Hypotheses 5 (H5).** *Compatibility has a direct effect on behavioral intention to use.*

Previous research has demonstrated that the TAM's predictive ability could be enhanced by incorporating additional variables [26]. More precisely, it has been shown that the concepts of perceived self-efficacy and facilitating conditions are significant determinants of behavioral intention to use new technologies. Perceived self-efficacy is a construct that explains how an individual assesses their own ability to perform a task suc-

cessfully [27]. Perceived self-efficacy is a key predictor of behavior because completing a job depends not only on a person's knowledge but also on the person's belief in their ability to complete it. Concerning LCBs' acceptance, perceived self-efficacy would indicate how NCS producers would perceive their ability, experience, skills, and expertise required for the use of AD technology integrated into the NCS process. As a result, perceived self-efficacy plays an important role in LCB acceptance, leading to the following hypothesis:

**Hypotheses 6 (H6).** *Perceived self-efficacy has a direct effect on behavioral intention to use.*

Facilitating conditions refer to an individual's belief that an adequate organizational and technical infrastructure level exists to support the system's use [28]. In the context of LCB, facilitating conditions include various approaches to meeting producers' needs, such as related training programs and workshops, technical consultants, and technical guidelines. Therefore, the following hypothesis was included in the model:

**Hypotheses 7 (H7).** *Facilitating conditions have a direct effect on behavioral intention to use.*

In this study, the TAM was integrated with the DIT and two additional constructs (perceived self-efficacy and facilitating conditions) to model acceptance of LCB by NCS producers. TAM provided the constructs of perceived usefulness, perceived ease of use, and behavioral intention to use. The DIT was used to elicit the concept of compatibility. The model also incorporates the other two external constructs, namely, perceived self-efficacy and facilitating conditions. The model used in this study is depicted in Figure 1; the dotted line in the figure denotes the study's scope, which is to predict Colombian NCS producers' behavioral intention to accept LCBs.

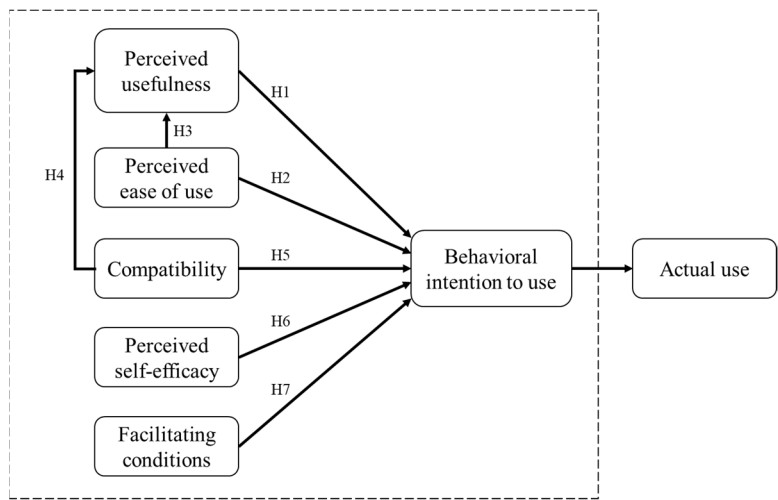

**Figure 1.** Extended technology acceptance model for LCB acceptance by NCS producers.

*2.2. Survey*

A structured survey was used to elicit data on the sample's sociodemographic characteristics, and a series of items was used to assess the constructs. The measures for all constructs were adapted from previously validated instruments and contextualized for LCB acceptance. Perceived usefulness, perceived ease of use, and behavioral intention to use were derived from prior research on the TAM [22,29,30]. The compatibility measures were adapted from Rogers [24], while the LCB perceived self-efficacy items were tailored from Venkatesh and Davis [29]. Venkatesh et al. [28] provided the facilitating conditions items modified for use in the current study.

The study's data collection method was through interviews with NCS producers. The survey consisted of two sections. The first section consisted of six questions capturing NCS producers' and production units' characteristics, including sex, age, formation, NCS production experience, sugarcane area, and yearly NCS production. The second section

collected data on producers' perceptions of the model's variables. This section included 16 items that assessed the six constructs (perceived usefulness, perceived ease of use, compatibility, perceived self-efficacy, facilitating conditions, and behavioral intention to use). All variables were measured using a five-point Likert scale (1: totally disagree, 2: disagree, 3: I have no idea, 4: agree, and 5: totally agree). The survey was initially reviewed by three NCS producers and three local extension agents to adjust the elements and constructs used in the research and explain the instrument's terminology, content, and general design. The survey application was then tested with 12 NCS producers to ensure they understood the questions, technical terms, and measurement scales. The researchers' observations and feedback from producers and local extension officers resulted in minor revisions to the survey instructions, the rewording of several items, and an explanation for several technical terms. Once the instrument was reviewed and adjusted, the definitive information was collected for the investigation.

### 2.3. Study Area and Sample

Figure 2 shows the area planted with sugar cane for the production of NCS in Colombia distributed in the country by departments. The following departments are highlighted: Boyacá, Santander, Nariño, Antioquia, Cundinamarca, Tolima, Huila, and Cauca. The annual average air temperature for cultivation is 26–32 °C during the day and 13–17 °C at night [31]. The AD technology-based alternative for waste management is still in its early stages of adoption, which is why this research is so important for the government and private entities seeking its early implementation.

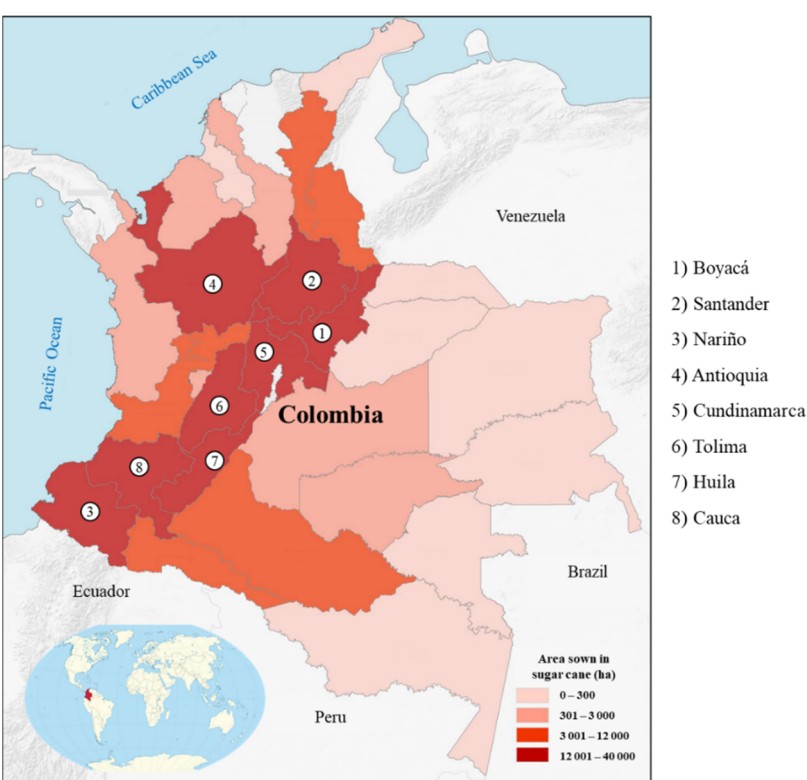

**Figure 2.** Map of the study area: sugarcane plantations in Colombia used to produce NCS; adapted from AGRONET [32].

Snowball and convenience sampling techniques were used to collect data for this study. Initially, respondents were chosen on the basis of proximity and ease of access to the researchers during the survey, using convenience sampling as a nonprobability sampling technique with an accidental sampling technique. The first wave of respon-

dents recommended potential NCS producers to include in this study via a chain-referral system [33].

### 2.4. Data Analysis

Linear structural relations (LISREL 10.20 [34]), a conventional and multilevel structural equation modeling program, was used for descriptive statistics and data modeling. The measurement instrument's reliability and validity were assessed using reliability, discriminant, and convergent validity criteria. Cronbach's alpha coefficient was used to determine the survey instrument's reliability [35]. Convergent and discriminant constructs validity were examined using average variance extracted analysis [36] and Pearson's correlation coefficient evaluated using Evans [37] guidelines (correlation levels: negligible = 0.00–0.19, weak = 0.20–0.39, moderate = 0.40–0.59, strong = 0.60–0.79, very strong = 0.80–1.00). The kurtosis and skewness of each construct were computed to verify the distribution of the data (normality). Additionally, exploratory factor analysis was used to determine the convergent validity of each construct.

The hypothesized relationships were tested using structural equation modeling. Path analysis was used to test the hypothesized relationships between variables and the theoretical model presented in Figure 1 based on multiple regression analyses. According to the model developed in the theoretical framework, two regression models were used to investigate the relationships between the variables. The relationship between perceived usefulness, the dependent variable, and compatibility and perceived ease of use, the independent variables, was examined in model 1. Model 2 examined the relationship between behavioral intention to use and perceived usefulness, perceived ease of use, compatibility, perceived self-efficacy, and facilitating conditions.

Evaluation of model fit is not as straightforward in structural equation modeling as in statistical approaches based on error-free variables. Because no single statistical significance test can reliably identify the correct model given the sample data, it is necessary to consider multiple criteria and evaluate model fit using multiple concurrent measures. For each estimation procedure, many goodness-of-fit indices are given to determine if the model is consistent with the empirical evidence. The present study used the significance test ($\chi^2$ test statistic) and descriptive goodness-of-fit measures to evaluate the model's fit. For the latter, the following indices were used [38]: root-mean-square error of approximation (RMSEA), standardized root-mean-square residual (SRMR), non-normed fit index (NNFI), comparative fit index (CFI), goodness-of-fit index (GFI), and adjusted goodness-of-fit index (AGFI).

## 3. Results and Discussion

### 3.1. Descriptive Statistics

In total, 187 responses were collected from Colombia's major NCS-producing departments. Five of the cases that responded provided insufficient information and were discarded, leaving 182 questionnaires completed. MacCallum [39] suggested 100–200 cases to obtain factor solutions that are adequately stable and closely correspond to the population factors.

Table 1 shows the demographic profile of the respondents. Agro-industrial activities of the processing of sugarcane to produce NCS have been led by the male gender. Women in rural areas of developing countries are frequently prevented from working outside the home and on family farms due to cultural, social, and religious norms [40]. The majority of respondents were in the age category 46–55 years. Meanwhile, almost 30% of the respondents corresponded to a population entering older adulthood (>56 years). This aging phenomenon is currently facing the NCS agro-industry, in which a slow generational change is perceived. Most NCS producers had a good education level; only 6.59% had no formal education at all, 39.01% had an elementary school education (5 years of schooling), 26.37% had completed high school (11 years of education), and some NCS producers (28.02%) had obtained a college degree (education spanning more than 11 years).

**Table 1.** Demographic attributes of the respondents (*N* = 182).

| Variables | Frequency | Percentage |
|---|---|---|
| Gender | | |
| Female | 13 | 7.14 |
| Male | 169 | 92.86 |
| Age (mean = 49.46 years) | | |
| Less than 35 years | 30 | 16.48 |
| From 35 to 45 years | 28 | 15.38 |
| From 46 to 55 years | 73 | 40.11 |
| From 56 to 65 years | 33 | 18.13 |
| More than 65 years | 18 | 9.89 |
| Education | | |
| No education at all | 12 | 6.59 |
| Elementary school | 71 | 39.01 |
| High school graduate | 48 | 26.37 |
| Some college | 51 | 28.02 |
| NCS production experience (mean = 29.14 years) | | |
| Less than 10 years | 18 | 9.89 |
| From 10 to 20 years | 41 | 22.53 |
| From 21 to 30 years | 15 | 8.24 |
| More than 30 years | 108 | 59.34 |
| Area sowed in sugar cane (mean = 21.68 ha) | | |
| Less than 5 ha | 13 | 7.14 |
| From 5 to 25 ha | 136 | 74.73 |
| From 26 to 50 ha | 25 | 13.74 |
| From 51 to 75 ha | 5 | 2.75 |
| More than 75 ha | 4 | 2.20 |
| Annual NCS production (mean = 150.72 t) | | |
| Less than 25 t | 12 | 6.59 |
| From 25 to 50 t | 35 | 19.23 |
| From 51 to 100 t | 59 | 32.42 |
| From 101 to 200 t | 45 | 24.73 |
| More than 200 t | 31 | 17.03 |
| NCS producer location | | |
| Boyacá | 25 | 13.74 |
| Santander | 27 | 14.84 |
| Nariño | 18 | 9.89 |
| Antioquia | 22 | 12.09 |
| Cundinamarca | 34 | 18.68 |
| Tolima | 20 | 10.99 |
| Huila | 22 | 12.09 |
| Cauca | 14 | 7.69 |

The majority of NCS producers surveyed had extensive experience producing NCS (average 29.14 years); a sizable percentage (59.34%) had more than 30 years of experience. Because NCS production has been a family tradition, the link with the agro-industry is established at an early age. The vast majority (74.73%) of NCS producers owned between 5 and 25 ha of sugarcane. As a result, the majority of producers in the study area were small-scale. NCS production averaged 150.72 t per year, with most producers producing between 51 and 100 t (sugarcane yield ranges between 4 and 11 t·year$^{-1}$·ha$^{-1}$). The location of the NCS producers revealed a dispersion across the Colombian territory, encompassing the primary NCS producing departments for this study.

### 3.2. Reliability and Validity Testing

The initial statistical results of the data collected with the measurement instrument are shown in Table 2.

**Table 2.** Statistical analysis of the data obtained with the measuring instrument.

| Construct | Compatibility | Perceived Ease of Use | Perceived Usefulness | Perceived Self-Efficacy | Facilitating Conditions | Behavioral Intention to Use |
|---|---|---|---|---|---|---|
| Covariance matrix | | | | | | |
| Compatibility | 0.84 | | | | | |
| Perceived ease of use | 0.28 | 0.62 | | | | |
| Perceived usefulness | 0.41 | 0.27 | 0.74 | | | |
| Perceived self-efficacy | 0.25 | 0.23 | 0.26 | 0.79 | | |
| Facilitating conditions | 0.32 | 0.22 | 0.33 | 0.33 | 0.88 | |
| Behavioral intention to use | 0.51 | 0.38 | 0.47 | 0.42 | 0.49 | 0.90 |
| Pearson's correlation matrix | | | | | | |
| Compatibility | 1.00 | | | | | |
| Perceived ease of use | 0.46 (0.000) | 1.00 | | | | |
| Perceived usefulness | 0.53 (0.000) | 0.50 (0.000) | 1.00 | | | |
| Perceived self-efficacy | 0.31 (0.004) | 0.41 (0.000) | 0.36 (0.000) | 1.00 | | |
| Facilitating conditions | 0.36 (0.000) | 0.36 (0.000) | 0.42 (0.000) | 0.40 (0.000) | 1.00 | |
| Behavioral intention to use | 0.62 (0.000) | 0.64 (0.000) | 0.63 (0.000) | 0.55 (0.000) | 0.58 (0.000) | 1.00 |
| Statistics | | | | | | |
| Mean | 4.19 | 4.57 | 4.41 | 4.12 | 4.08 | 4.30 |
| Standard deviation | 0.92 | 0.65 | 0.82 | 0.97 | 0.97 | 0.90 |
| Skewness | 1.02 | 1.36 | 1.30 | 1.09 | 1.26 | 1.22 |
| Kurtosis | 0.45 | 1.22 | 1.05 | 1.13 | 1.38 | 0.92 |
| Cronbach's alpha | 0.85 | 0.93 | 0.83 | 0.83 | 0.89 | - |

The average variance extracted (AVE) square root is presented on the covariance matrix's main diagonal. The values in parentheses in the Pearson correlation matrix show the significance of the values (two-tailed).

Cronbach's alpha coefficient of the measurement instrument had an average value of 0.87, indicating a high internal consistency level, since it was higher than the recommended minimum of 0.8 for basic research purposes [34]. Therefore, the reliability of the survey instrument was confirmed. Furthermore, the average variance extracted (AVE) square root was much larger than all other cross-correlations for the sample. The square root of the AVE for all measures exceeded the recommended level of 0.5 [35] (ranged from 0.62 to 0.90), indicating that the hypothesized constructs accounted for more than half of the variability observed in the items. Pearson's correlation matrix results showed that the independent variables were positively correlated from moderate to strong with behavioral intention to use (ranged from 0.55 to 0.64). In contrast, perceived usefulness was linked from weak to moderate with compatibility and perceived ease of use. However, all variables were significantly correlated with the behavioral intention to use and each other at $p < 0.01$ (values in parentheses). The mean values of all variables were above four and with standard deviations between 0.65 and 0.97, indicating that the vast majority of respondents agreed with or tended to agree with the variables' statements. Furthermore, the distribution of the data was mainly of moderate normality since the absolute values of skewness and kurtosis averaged 1.18, which is in the range of 1 to 2.3 reported by Lei and Lomax [41], except for compatibility kurtosis, which was considered a normal distribution (<1). As a result, the data analysis using structural equation modeling was adequate.

Constructs with the measurable indicators and the factor loading are shown in Table 3. The factor loading coefficient is the correlation coefficient between the constructs and the

factor analysis measures. This analysis revealed that all measures had factor loadings greater than 0.6 (if a correlation with an instrument measuring the same construct is >0.50, convergent validity is generally considered adequate [42]), thus verifying the convergent validity of each construct. Thus, the criteria above confirmed the measurement instrument's reliability, convergent validity, and discriminant validity.

**Table 3.** Factor analysis between the constructs and the measurable variables.

| Construct | Measures | Factor Loading |
|---|---|---|
| Compatibility | Using the low-cost biodigesters is compatible with most aspects of an NCS mill | 0.83 |
| | Using low-cost biodigesters to produce bioenergy and biofertilizer is compatible with the environment and climate of this region | 0.89 |
| | Using the low-cost biodigesters for the benefit of NCS production is consistent with the financial situation of the process | 0.71 |
| Perceived ease of use | Learning to operate the low-cost biodigesters would be easy for me | 0.94 |
| | The interaction with the low-cost biodigesters would be easy for me to understand | 0.86 |
| | I would find the low-cost biodigesters easy to use | 0.91 |
| Perceived usefulness | Using the low-cost biodigesters would save time and money | 0.72 |
| | The low-cost biodigesters would support critical aspects in an NCS mill | 0.82 |
| | I would find the low-cost biodigesters useful in an NCS mill | 0.81 |
| Perceived self-efficacy | I could use the low-cost biodigesters if there were no one around to tell me what to do as I go | 0.64 |
| | I could use the low-cost biodigesters if I saw someone else using them before trying them myself | 0.86 |
| | I could use the low-cost biodigesters if someone showed me how to do it first | 0.90 |
| Facilitating conditions | I have the resources necessary to use the low-cost biodigesters | 0.88 |
| | I have enough knowledge to use the low-cost biodigesters | 0.84 |
| | Given the resources, opportunities, and knowledge it takes to use the low-cost biodigesters, it would be easy for me to use it | 0.83 |
| Behavioral intention to use | Assuming I had access to the low-cost biodigesters, I would intend to use it | - |

The experience of the producers in NCS production confirmed the maturity of the studied sector (Table 1). Likewise, the low level of non-schooling, associated with the acceptance of the variables investigated, allowed us to discern a concern among NCS producers based not only on survival but also on environmental conservation. Therefore, producers perceived that AD technology could be a solution for sustainable waste management and agro-industry benefit.

*3.3. Model Fit Evaluation*

The maximum likelihood was the fit function used for the structural equation models. It is consistent and efficient, does not depend on the scale, and is normally distributed if the observed variables are moderately normal [43]. The extent to which the specified models fit the empirical data is shown in Table 4. The degrees of freedom (df) were 24 and 90 for models 1 and 2, respectively. Therefore, the $\chi^2$ test statistic associated with the significance test (*p*-value) demonstrated the good fit of the models to the data.

The root-mean-square error of approximation (RMSEA) is a statistic that indicates the population's approximate fit and, hence, the difference caused by approximation [44]. RMSEA values were zero in both models; thus, it was inferred that they fit the population approximately well. Furthermore, the lower boundary (left side) of the 90% confidence interval for RMSEA was zero for both models. The standardized root-mean-square residual index (SRMR) is an overall badness-of-fit measure based on the fitted residuals first divided by the standard deviations [45]. Models were found to have an SRMR within the good model's fit range. The other descriptive measures used to evaluate the fit of the models (NNFI, CFI, GFI, and AGFI) presented good fit values according to the literature reviewed.

Considering the previous results on the statistical indices, the adjustment of the proposed models was validated.

**Table 4.** Results of the model fit evaluation.

| Fit Measure | Good Fit | Reference | Model 1 | Model 2 |
|:---:|:---:|:---:|:---:|:---:|
| $\chi^2$ $p$-value | $0 \le \chi^2 \le 2df$ $0.05 < p \le 1.00$ | [46] | 17.31 0.84 | 79.37 0.78 |
| RMSEA | $0 \le \text{RMSEA} \le 0.05$ | [44] | 0.00 | 0.00 |
| SRMR | $0 \le \text{SRMR} \le 0.05$ | [45] | 0.02 | 0.03 |
| NNFI | $0.97 \le \text{NNFI} \le 1.00$ | [47] | 1.00 | 1.00 |
| CFI | $0.97 \le \text{CFI} \le 1.00$ | [48] | 1.00 | 1.00 |
| GFI | $0.95 \le \text{GFI} \le 1.00$ | [49] | 0.98 | 0.95 |
| AGFI | $0.90 \le \text{GFI} \le 1.00$ close to GFI | [50] | 0.96 | 0.92 |

$\chi^2$: chi-square. *df*: degrees of freedom (model 1 = 24, model 2 = 90). RMSEA: root-mean-square error of approximation. SRMR: standardized root-mean-square residual. NNFI: non-normed fit index. CFI: comparative fit index. GFI: goodness-of-fit index. AGFI: adjusted goodness-of-fit index.

*3.4. Hypothesis Testing*

The results of structural equation modeling are shown in Table 5. The unstandardized coefficients of the independent variables showed positive correlations with their dependent variables. The standard error averaged 0.073 and 0.111 for the independent variables in models 1 and 2, respectively, which indicates an appropriate estimate. In models 1 and 2, the standard error was lower (on average, 0.0215). The standard error shows how precisely the parameter's value was estimated; a smaller standard error denotes a more accurate estimate. Furthermore, models 1 and 2 had variance errors of 0.0826 and 0.180, respectively, sufficient for establishing the parameters. From the Z-values, it is possible to reject the null hypothesis and accept the hypotheses proposed for each model (Z-values greater than 1.96 at a significance level of 5%).

**Table 5.** Multiple regression analysis results.

| Independent Variable | Unstandardized Coefficients | | Standardized Coefficients | Z-Values | *p*-Values |
|:---:|:---:|:---:|:---:|:---:|:---:|
| | β | Standard Error | β | | |
| Model 1: | Dependent variable: Perceived usefulness | | | | |
| Model statistics | Errorvar = 0.0826, $R^2$ = 0.741, Standard error = 0.0214, Z-value = 3.84, *p*-value = 0.00 | | | | |
| Compatibility | 0.48 | 0.072 | 0.64 | 6.66 | 0.000 |
| Perceived ease of use | 0.27 | 0.073 | 0.29 | 3.68 | 0.000 |
| Model 2: | Dependent variable: Behavioral intention to use | | | | |
| Model statistics | Errorvar = 0.180, $R^2$ = 0.777, Standard error = 0.0216, Z-value = 8.32, *p*-value = 0.00 | | | | |
| Compatibility | 0.25 | 0.112 | 0.21 | 2.23 | 0.026 |
| Perceived ease of use | 0.26 | 0.093 | 0.18 | 2.82 | 0.005 |
| Perceived usefulness | 0.41 | 0.189 | 0.26 | 2.19 | 0.029 |
| Perceived self-efficacy | 0.28 | 0.0877 | 0.19 | 3.14 | 0.002 |
| Facilitating conditions | 0.25 | 0.0721 | 0.21 | 3.41 | 0.001 |

The combined hypotheses' path analysis results are shown in Figure 3. Compatibility and perceived ease of use accounted for 74% of the variance in perceived usefulness in model 1. Additionally, compatibility had the greatest effect on perceived usefulness (H4) and was strongly supported (β = 0.64, Z = 6.66, *p* < 0.001). On the other hand, H3 achieved a lower β compared to H4. However, its contribution was also significantly supported (β = 0.29, Z = 3.68, *p* < 0.001). Compatibility, perceived ease of use, perceived

usefulness, perceived self-efficacy, and facilitating conditions explained 78% of the variance of behavioral intention to use (model 2).

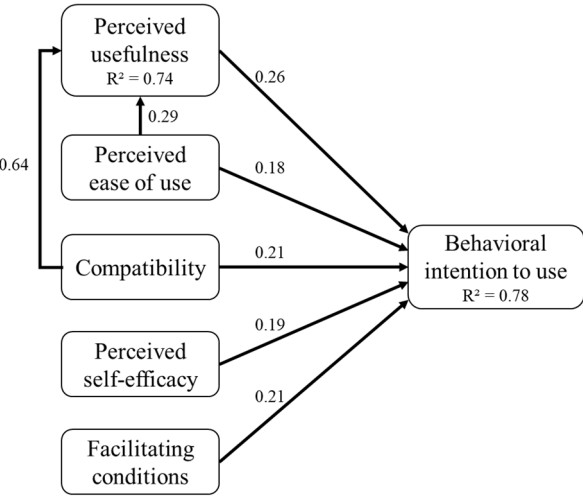

**Figure 3.** Illustration of the study's empirical findings following the proposed model.

In model 2, the proposed Hypotheses H2 and H5 were corroborated, indicating that perceived ease of use and compatibility also directly affect behavioral intention to use. In this model, compatibility obtained a β greater than that of perceived ease of use (0.21 > 0.18), but both effects (H2 and H5) were supported ($Z > 1.96$, $p < 0.05$). Similarly, the effect of perceived usefulness on behavioral intention to use (H1) was verified (β = 0.26, $Z = 2.19$, $p < 0.05$), obtaining the highest β coefficient among the proposed constructs on behavioral intention to use. H6 examined the relationship between perceived self-efficacy and behavioral intention to use. As expected, there was a significant correlation (β = 0.19, $Z = 3.14$, $p < 0.01$), confirming this hypothesis. H7 was associated with the effect of facilitating conditions on behavioral intention to use. H7's path was significant (β = 0.21, $Z = 3.41$, $p < 0.01$). Therefore, H7 was supported. Overall, it was determined that all path coefficients were statistically significant.

There were indirect effects of some explanatory variables on the dependent variables, in addition to the direct effects summarized in Figure 3. These effects resulted from multiplying all of the direct effects from the explanatory variable through the causal path (e.g., compatibility on perceived usefulness) to the final dependent variable (e.g., behavioral intention to use). Table 6 summarizes the total effect of each explanatory variable on the dependent variables, including direct and indirect effects. The strongest overall effect was for compatibility on the behavioral intention to use LCB (0.38), followed by perceived ease of use and perceived usefulness (both 0.26), facilitating conditions (0.21), and perceived self-efficacy (0.19).

**Table 6.** Variables' direct, indirect, and total effects on dependent variables.

| | Effect on | | | | | |
| --- | --- | --- | --- | --- | --- | --- |
| **Variable** | **Perceived Usefulness** | | | **Behavioral Intention to Use** | | |
| | **Direct** | **Indirect** | **Total** | **Direct** | **Indirect** | **Total** |
| Compatibility | 0.64 | - | 0.64 | 0.21 | 0.17 | 0.38 |
| Perceived ease of use | 0.29 | - | 0.29 | 0.18 | 0.08 | 0.26 |
| Perceived usefulness | | | | 0.26 | - | 0.26 |
| Perceived self-efficacy | | | | 0.19 | - | 0.19 |
| Facilitating conditions | | | | 0.21 | - | 0.21 |

The total effect is equal to the sum of the direct and indirect effects. The direct effects occur without the intervention of any other variable in the model (i.e., all significant beta values from Figure 3). The indirect effects were calculated by multiplying the coefficients of each independent variable by the coefficients of the related dependent variables.

The extended TAM developed in this study was successfully applied in the context of LCB acceptance by NCS producers, demonstrating the model's robustness. The results indicate that an extended TAM model can capture some of the unique contextual characteristics of NCS producers in terms of LCB acceptance. The findings suggest that producers' intentions accurately predict the factors that affect the acceptance of AD technology by NCS producers, confirmed by pathway analysis (Figure 3), showing that all seven initial hypotheses were supported (Tables 5 and 6). Thus, behavioral intention is significantly affected by compatibility, perceived ease of use, perceived usefulness, perceived self-efficacy, and facilitating conditions.

Similar findings have been reported in previous studies examining the role of farmers' intentions in accepting new technologies through TAM. For instance, when analyzing the intention of sugar beet farmers to accept drip irrigation using the TAM with eight constructs [21], it was possible to account for 59.3% of the variation in farmers' behavioral intentions. In another study, when explaining the acceptance and use of biological control for rice stem borer [16], the TAM explained 78% of the variance in the behavioral intention; however, the perceived ease of use construct was not significant. In contrast, this construct was substantial in the present study, and the two-part model explained 74% of the perceived usefulness variance and accounted for 78% of behavioral intention to use.

The TAM is a well-known technology acceptance theory, and evidence suggests it holds in the agricultural context. However, a new model must be developed for each sector to explain why a specific population accepts a certain technology; hence, the results of this work confirm its importance.

The most significant effect on behavioral intention to use was compatibility (H5). This result corroborates other hypotheses stating the critical role of compatibility with other innovations to explain why the new technology is adopted [22,24]. Compatibility has been crucial in selecting an appropriate rice stem borer management [51] and adopting precision agriculture technology [15]. Compatibility may have had a high priority in the current study because LCB is an on-site waste management technology for bioenergy and biofertilizer production tailored to the needs of the NCS producer. Producers' understanding that LCB is consistent with the farm's environmental conditions, process needs, and affordability of the technology allows them to have a high perception of the advantages of LCB. As a result, they will be more receptive to using LCB on their farms. Compatibility appears to be the most important determinant of LCB success, and, as such, it must be considered when promoting and implementing a technology transfer program. On the other hand, several research studies have found that compatibility directly impacts perceived usefulness [16,52]. By knowing the advantages of LCB in advance, producers are more likely to consider the usefulness of LCB if they consider it compatible with most of their farm conditions and NCS processing activities (supporting H4).

In previous studies [23], perceived usefulness was consistently a strong predictor of usage intention, and similar findings were found in this study (H1). Overall, respondents found LCB useful for waste management, even though the technology has yet to be implemented. Currently, LCB is a useful solution for organic waste generated in various neighboring agro-industries that have benefited economically and environmentally from the technology [53,54]. Perhaps due to these personal experiences, NCS producers are more receptive to technology acceptance, as the utility benefits of LCB outweigh the disadvantages of traditional waste management methods such as landfills or land disposal.

Like the perceived usefulness, the perceived ease of use, as one of the main constructs of the original TAM, showed a strong determinant on the behavioral intention of use (H2). Perceived ease of use refers to the degree to which an NCS producer "feels and perceives" the use of an LCB effortlessly. Previous research has indicated that improving perceived ease of use increases producers' readiness to accept a new technology [21]. Additionally, in the pre-implementation test, perceived ease of use directly and significantly affected behavioral intention to use (little or no direct experience with a particular system) [26]. In the present study, the producers perceived the use of LCB as easy. Therefore, perceived

ease of use is a critical variable to consider during the early stages of dissemination, such as the current study, because the success of technological diffusion is highly dependent on it. Consistent with previous research findings [55,56], it was found that perceived ease of use had a substantial positive effect on the perceived utility of LCB implementation (supporting H3). This result indicates that if NCS producers can easily use the LCB, they will consider it more useful.

Another novel result is the effect of facilitating conditions on producers' behavioral intention to use LCB in their practices (H7). Facilitating conditions represent a relatively recent concept used in technology acceptance studies, but they are critical for potential research applications [16]. The findings for this factor support the view that NCS producers who frequently receive extension services and training would welcome any opportunity to promote the use of LCB on their farms and support efforts to disseminate this technology.

The concept of perceived self-efficacy is central to social learning theory. As an individual factor, perceived self-efficacy reflects an individual's beliefs about their ability to complete specific tasks [27]. In the context of the NCS agro-industry, individuals' perceptions of their knowledge, skill, and capability for LCB application are reflected in their perceived self-efficacy. According to a systematic literature review of TAM studies, perceived self-efficacy is the first most commonly used external construct [57]. In this respect, the role of perceived self-efficacy is key to understanding the behavioral intention of NCS producers to use LCB (supporting H6). This finding could be explained by the fact that people with high self-efficacy believe they can perform well using technology [58]. As a result, they are more likely to try the technology and continue to evaluate its benefits.

## 4. Conclusions

Anaerobic digestion is considered the future of renewable energies with a sustainable approach. In this sense, social acceptance must be analyzed to disseminate the LCB without barriers in agricultural sectors as NCS producers. In this study, the extended technology acceptance model (TAM) predicted the behavioral intention of non-centrifugal cane sugar (NCS) producers to use low-cost digesters (LCB) in Colombia. The TAM was successfully extended (in terms of model fit) by including three external factors relevant to analyze the acceptance of LCB: perceived self-efficacy, facilitating conditions, and compatibility as antecedents of behavioral intention to use. Path analysis showed that the seven hypotheses proposed for the extended TAM were adequately supported with $Z$-values greater than 1.96 at a significance level of 5%. The model's constructs explained 74% of the variance in perceived usefulness and 78% in behavioral intention to use LCB. The unexplained variance of the dependent variable implies that other variables can also be found within the framework to increase the explanation level. For example, subjective norms and personal relationships that seem to be about behavioral intention toward the acceptance of LCB can be suitable determinants. In this work, the behavioral intention was explained with five factors (i.e., compatibility, perceived self-efficacy, facilitating conditions, perceived usefulness, and perceived ease of use). However, additional factors, such as demographic characteristics of producers (e.g., gender, age, and education) and farm structure characteristics (land size, sugarcane yield, etc.), also exist. These additional factors can also influence the behavioral intention to use the AD technology. Considering this perspective, future research needs to include such elements to build a comprehensive model while maintaining conciseness. Lastly, Colombian NCS producers tended to concur with accepting the integration of anaerobic digestion (AD) technology with the NCS agro-industry. There is a sense of urgency surrounding implementing these AD systems in the NCS agro-industry, which would avoid significant environmental impacts while also generating economic benefits and social inclusion. The findings of this study contribute to a new understanding of NCS producers' perspectives on AD and serve as a guide for developing strategies and resource management for developing countries' technology diffusion policies.



**Author Contributions:** Conceptualization, O.M. and E.V.; methodology, O.M.; software, E.V.; validation, E.V.; formal analysis, O.M.; investigation, O.M.; resources, J.R.; data curation, E.V.; writing—original draft preparation, L.C.; writing—review and editing, O.M.; visualization, O.M.; supervision, H.E.; project administration, J.R.; funding acquisition, L.C. All authors have read and agreed to the published version of the manuscript.

**Funding:** This research received no external funding.

**Institutional Review Board Statement:** Not applicable.

**Informed Consent Statement:** Not applicable.

**Acknowledgments:** The authors would like to acknowledge the support of the Ministerio de Ciencia Tecnología e Innovación (Minciencias) convocatoria 757 de 2016, Corporación Colombiana de Investigación Agropecuaria (AGROSAVIA), Federación Nacional de Productores de Panela (FEDE-PANELA), Instituto Técnico Agrícola de Cáchira (Norte de Santander-Colombia), and Universidad Industrial de Santander (UIS) for the development of this work.

**Conflicts of Interest:** The authors declare no conflict of interest. The funders had no role in the design of the study; in the collection, analyses, or interpretation of data; in the writing of the manuscript, or in the decision to publish the results.

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
