# Peer review of "Toward the Adoption of Anaerobic Digestion Technology through Low-Cost Biodigesters: A Case Study of Non-Centrifugal Cane Sugar Producers in Colombia"

_water, doi:10.3390/w13182566_

Round 1

Reviewer 1 Report

Abstract

Abbreviations should not be present. Hence, it is suggested to use the extended form.

TAM appears for the first time, without having the extended form.

Introduction

The introduction is specifically centred on the method for managing/research of condition for LCB acceptance.

The reviewer suggests to give an appropriate introduction about the anaerobic digestion, the anaerobic digestion aimed at processing cane sugar (since its waste constitutes a significant amount in the associate agro-industry field, as  reported in lines 180 - 184), therefore, also giving an overview on the biodigesters technology.

The anaerobic digestion technology is much centred in the title, therefore the reader expects of having a significant overview on it, specifically focused on the efficient and sustainable energy. As sometimes written, the field of application is that of the circular bio-energy and bio-economy. Hence, the authors should be deepen this aspect and well frame as the LCB processing cane sugar waste in Colombia can contribute, also in term of expected energy and environmental impact.

In this regards, it would be opportune to give a spot on inntegrated systems, based on anaerobic digestion tecnology as primary biofuel producer.

Following some bibliography suggestions are listed for this field.

https://doi.org/10.1016/j.fuel.2017.01.106

https://doi.org/10.1080/15567036.2011.592908

https://www.aimspress.com/article/doi/10.3934/energy.2021044

https://doi.org/10.1016/j.resconrec.2021.105832

The authors must write well on the distintive points of their manuscript, and on the contribution of it in the research field.

Line 169 – 170. The pilot study comprises 12 NCS producers. The question are the following. Was this a preliminary test before conducting the real survey? Or do the 12 NCS producers represent the definitive basin data for your analysis?

This is not clear, and could be in contrast with written in line 227.

Line 195. Describe LISREL.

Lines 219 – 224. The reviewer opinion is that of showing the core equations that are the base for accomplishing the scientific considerations, although they are known.

Results and discussion

Descriptive statistics

The authors should emphasize on what attributes of the respondents are more important than others (trying to have the accent on the specific characcteristics of same respondents, as for example expertise, experience acquired in the field, education, etc.). This has an impact on the following paragraph “reliability and validity testing”.

Hypothesis testing

Lines 360 – 364. The considerations reported in the lines must be discussed deeply. “The findings suggest .. of LCB acceptance” this conclusion must be accompanied by a number window, within it the “acceptance” can be declared. Moreover, it is opportune to compare the current findigs to those of refs. [10, 13].

Conclusion

The conclusion paragraph must be well framed with more numbers, and more centred in scientific terms.

Reviewer 2 Report

The title is clear.

The content is in accord with title.

The Editors must establish if the manuscript is in journal topics.

The manuscript adheres to the journal's standards after revision.

The size of the article is appropriate to the contents.

The authors must underline the major findings of their work and explain novelty of this study comparatively with their published papers or other similar studies. The novelty must be better pointed.

The Abstract must be revised.

The key words permit found article in the current registers or indexes, but these can be revised for example: behavioral intention;

In the introduction it isn’t clearly described the state of the art of the investigated problem. The references from the last years (2020 and 2021) are necessary. Is this study actual?

The paper was written in standard, grammatically correct English, small corrections are necessary.

The figures have a good quality.

The tables contain necessary results.

The Conclusion must be revised.

References from last years are necessary.

Please provide minimum 2 references from this journal (2021), for demonstrated that manuscript is in Water topic.

The paper has the text presented and arranged clearly and concisely.

Please respect guide for authors.

Please verify references and respect guide for authors.

Round 2

Reviewer 1 Report

In my opinion the issue of the circular energy, involving this important field as the waste to energy from organics should have been described well. It would have been opportune to give an overview on the current major energy technologies matching the gasifiers, and a spot on those innovative, on which the research is interested for a close future, in view of efficiency and sustainability. Still remaining that some suggestions can be better acquired by the authors, the manuscript can be accepted in this present form. 

Reviewer 2 Report

The manuscript was improved.